# Conjunctival Injection Reduction in Patients with Atopic Keratoconjunctivitis Due to Synergic Effect of Bovine Enteric-Coated Lactoferrin in 0.1% Tacrolimus Ophthalmic Suspension

**DOI:** 10.3390/jcm9103093

**Published:** 2020-09-25

**Authors:** Hiroshi Fujishima, Naoko Okada, Kenji Matsumoto, Eisuke Shimizu, Shinji Fukuda, Masaru Tomita

**Affiliations:** 1Department of Ophthalmology, Tsurumi University School of Dental Medicine, Tsurumi, Yokohama 230-8501, Japan; n-okada@nichiyaku.ac.jp; 2Department of Pharmaceutical Sciences, Nihon Pharmaceutical University, Saitama 362-0806, Japan; 3Department of Allergy and Clinical Immunology, National Research Institute for Child Health and Development, Tokyo 157-8535, Japan; matsumoto-k@ncchd.go.jp; 4Department of Ophthalmology, Keio University School of Medicine, Tokyo 160-8582, Japan; ophthalmolog1st.acek39@keio.jp; 5Institute for Advanced Biosciences, Keio University, Tsuruoka, Yamagata 997-0052, Japan; sfukuda@sfc.keio.ac.jp (S.F.); mt@sfc.keio.ac.jp (M.T.); 6Intestinal Microbiota Project, Kanagawa Institute of Industrial Science and Technology, Kawasaki 210-0821, Kanagawa, Japan; 7Transborder Medical Research Center, University of Tsukuba, Tsukuba 305-8575, Ibaraki, Japan

**Keywords:** lactoferrin, tacrolimus ophthalmic suspension, conjunctival injection, atopic keratoconjunctivitis

## Abstract

Lactoferrin (LF), a multifunctional glycoprotein found in mammalian milk, is reported to have immunoregulatory effects. The present study aimed to evaluate whether enteric-coated LF (eLF) could improve symptoms in patients with atopic keratoconjunctivitis (AKC). This randomized double-blind placebo-controlled single-center trial comprised Japanese patients (n = 20; aged 22–60 years) with AKC. Patients treated with 0.1% tacrolimus ophthalmic suspension (TALYMUS^®^) were administered eLF (400 mg/d of bovine LF) or placebo tablets for 12 weeks. Conjunctival injection was examined, papillae formation in the palpebral conjunctiva was evaluated, and corneal fluorescein score, itchy sensation in end-point itching scale, and serum allergic parameters were assessed. Conjunctival injection was significantly reduced in the LF group than in the placebo group (*p* = 0.0017, Mann–Whitney U-test). Papillae formation in the palpebral conjunctiva showed a statistical decrease in the LF group than in the placebo group (*p* = 0.010, unpaired T-test). LF combined with TALYMUS^®^ could be a promising treatment strategy to mitigate AKC.

## 1. Introduction

Lactoferrin (LF) is a multifunctional iron-binding glycoprotein found at the highest concentrations in breast milk and has anti-bacterial, antiviral, immunostimulatory, antioxidant, and cancer-preventive potentials [1,2,3].

LF, which is present at high concentration in mucosal secretions, plays a critical role during non-specific defense against infections and inflammation [4]. Serum LF is produced by acinar cells in the lacrimal gland and possibly by tear neutrophils during infection and inflammation. By binding iron, LF inhibits iron-dependent pathogen growth [5], moreover, oral administration of bovine LF stimulates both systemic and mucosal immune responses in vivo [6].

LF has been identified in tears and vitreous humor, while its genes have been detected in human cornea and retinal pigment epithelium (RPE) cell cultures. Positive antibody reactions for human LF in the cornea, iris, and RPE tissues have been previously reported [7]. Oxidative stress induces ocular surface inflammation; LF present in tear film exhibits anti-microbial and anti-oxidative properties [7,8], making it a potential biomarker for mucosal immune competence [6].

Since LF is a natural component of breast milk, which is ingested by infants, it has been approved as a food additive in Japan, and is included in the “generally recognized as safe” category in the United States of America.

Atopic keratoconjunctivitis (AKC), associated with chronic and severe clinical courses, can be accompanied by conjunctival injection, corneal changes, tissue remodeling, and fibrosis, such as corneal ulcers and the formation of giant papilla, eventually leading to loss of vision [9,10]. When patients with AKC are refractory to first-line agents such as antihistamines, prolonged use of either topical or systemic corticosteroids or immune-suppressants is required. Therefore, development of effective therapeutic strategies to prevent serious comorbidities would be of utmost importance. However, how conjunctival and corneal changes are generated in chronic ocular allergic diseases, such as AKC, remains to be elucidated.

Immunosuppressants, such as cyclosporin ophthalmic suspension [11] and tacrolimus ointment [12,13] have been reported to mitigate symptoms in patients with AKC and reduce the rate of serious side effects due to steroids [14]. Tacrolimus, isolated from *Streptomyces tsukubaensis*, inhibits calcineurin activity. Approximately 0.1% tacrolimus ophthalmic suspension (TALYMUS^®^, Senju Pharmaceutical Co., Ltd., Osaka, Japan) has been recommended for treating several ocular diseases, including AKC and VKC, however, patients did not fully recover, with mild symptoms remaining. A previous report had indicated that not only pollen and animal dander, but higher levels of ambient pollutants and temperature, and lower humidity may also exacerbate allergic symptoms. Thus, additional therapeutic trials, besides 0.1% tacrolimus ophthalmic suspension, are warranted. Herein, we explored ways to enhance therapeutic efficacy of 0.1% tacrolimus ophthalmic suspension in the treatment of severe AKC throughout the year.

Lactate dehydrogenase (LDH), immunoglobulin E (IgE), thymus and activation-regulated chemokine (TARC), eosinophil number, and eosinophilic cationic protein (ECP) mediate allergic inflammation, and are produced in the serum of patients with allergic diseases, besides interleukin (IL)-4 and IL-13, the signature type 2 cytokines. These activities are important for the maintenance of allergic inflammation in atopic dermatitis or bronchial asthma. Eye lid elasticity and redness are also symptoms of AKC [15].

The aim of this randomized double-blind placebo-controlled single-center trial was to determine whether the synergic effect of enteric-coated LF (eLF) with TALYMUS^®^ could improve conjunctivitis in patients with AKC. We assessed the changes in conjunctival injection, serum protein levels, and eyelid skin conditions, following a 12-week treatment, with or without eLF.

## 2. Materials and Methods

### 2.1. Study Design

The treatment duration of this randomized double-blind placebo-controlled single-center trial was 12 weeks. Patients were allocated to one of the two groups, namely, the eLF group (daily ingestion of four eLF tablets; 400 mg/d of bovine LF) or the placebo group (daily ingestion of four enteric-coated placebo tablets). Patients consumed four treatment tablets per day for twelve weeks. Although there was no specific time for treatment, we recommended all subjects to take two tablets after breakfast, and two before sleep, to maintain compliance. Moreover, all subjects were instructed to maintain their usual intake and physical activity, and to re-visit for follow up at 4-week intervals throughout the study. This trial was performed between July 2014 and April 2015. The protocol was approved by the Tsurumi University Institutional Review Board (1209) and the trial was conducted in accordance with the tenets of the Helsinki Declaration, under the supervision of clinical investigators. Subjects provided informed consent, including permission for publication.

### 2.2. Test Material

Since orally administered proteins are generally degraded by pepsin, in the stomach, we used eLF tablets as the test material. The enteric-coated LF and placebo supplements were provided by Lion Co. (Kanagawa, Japan). eLF tablets contained 100 mg LF/tablet while the placebo enteric-coated tablets contained lactose instead of LF. Other constituents included crystalline cellulose, carboxymethylcellulose-Ca, sucrose ester, silicon dioxide, shellac, sorbitol, arginine, dextrin, and long pepper powder. In this formulation, LF molecules were protected from proteolytic digestion due to their coating with an acid-resistant material, shellac, which dissolves easily at neutral pH in the intestine. The characteristics of this formulation were assessed by the standard disintegration test to satisfy the Japanese Pharmacopoeia criteria.

### 2.3. Patients

Patients with AKC (15 men and 5 women; eLF group = nine patients, mean age: 30.0 years; placebo group = 11 patients, mean age: 28.1 years) were enrolled by the Department of Ophthalmology, Tsurumi University School of Dental Medicine in Yokohama, Japan. AKC was diagnosed in accordance with the Japanese guidelines for allergic conjunctival diseases [10], characterized by the following signs and symptoms: ocular itching, redness, tearing or pain, discharge, chemosis, hyperemia or papillae of the palpebral conjunctiva upon slit-lamp examination, medical history, and positivity for serum antigen-specific IgE, as indicated by SRL Inc. (Tokyo, Japan), using the multiple antigen simultaneous test (MAST Immunosystems Inc., Mountain View, CA, USA). All patients with AKC were topically treated with TALYMUS^®^ at least for a year prior to enrolment in the study. Treatments were not altered throughout the study period. Other systemic treatments, such as oral steroids or antihistamines and steroid or tacrolimus skin ointments, were not prescribed before or during the study period.

### 2.4. Evaluation Methods

#### 2.4.1. Serum Analysis

The serum concentrations of LDH, IgE, TARC, and ECP were determined by SRL Inc., before and after treatment, using a commercially available enzyme-linked immunosorbent assay. Serum eosinophils were also enumerated.

#### 2.4.2. Ocular Surface Evaluation

Three blinded allergy specialist-ophthalmologists, who were unaware of the study design, evaluated conjunctival injection (hyperemia of bulbar conjunctiva) of palpebral conjunctiva in both eyes, using a photographic biomicroscope. Grading, in both eyes, was performed according to a four-point scale: none (0), mild (1), moderate (2), and severe (3). Papillae formation in the palpebral conjunctiva was evaluated using the following clinical evaluation criteria for allergic conjunctival diseases: diameter > 0.6 mm (3), diameter 0.3–0.5 mm (2), diameter 0.1–0.2 mm (1), and no manifestation (0). Corneal fluorescein staining was graded as either shield ulcer or epithelial erosion (3), superficial punctate keratitis with filamentary debris (2), superficial punctate keratitis (1), or no manifestation (0).

The itchy sensation of pre- and end-point itching scale was scored from 0 to 4 points (0, none; 1, intermittent; 2, continuous; 3, severe itching; 4, incapacitating itching). Itching was enquired and scored accordingly by a doctor Hiroshi Fujishima (HF).

### 2.5. Sample Size and Statistical Analysis

Sample size was calculated based on the visceral fat accumulation as previously reported [16]. Twenty evaluable patients were required to achieve an 86% power with two-sided α = 0.05 to show a statistically significant difference between the two treatments.

## 3. Results

Conjunctival injection was significantly reduced in the LF with TALYMUS^®^ group compared to that in the placebo group following 12 weeks of treatment (Figure 1, *p* = 0.0017, Mann–Whitney U-test). The ophthalmologists had assigned 0, 1, 2, and 3 scores in both eyes of an LF with TALYMUS^®^-treated patient. Conjunctival papillae formation in the palpebral conjunctiva was also scored as 0, 1, 2, and 3, showing a significant decrease in the LF with TALYMUS^®^ group than in the placebo group (*p* = 0.010, unpaired T-test) (Figure 2). There was no significant change in the itchy sensation (*p* = 0.24) and corneal fluorescein score (*p* = 0.17) (Table 1).

Conjunctival injection and papillae formation in the palpebral conjunctiva were significantly reduced in the LF with TALYMUS^®^ group, although there was no significant change in itchy sensation and corneal fluorescein score. SD, standard deviation.

There was no significant difference in the serum concentrations of LDH, IgE, TARC, and ECP, before and after treatment (data not shown). No adverse effects were observed during the study period, or even 6 months following study completion.

## 4. Discussion

AKC causes conjunctival alterations with/without corneal involvement, leading to visual morbidity as well as VKC. The aim of this 12-week randomized double-blind placebo-controlled trial was to determine whether LF (400 mg/d) combined with TALYMUS^®^ could improve conjunctival injection caused by atopic immune reactions, in patients with AKC. Our results showed that the synergic effect of LF significantly reduced conjunctival injection and papillae formation compared with placebo. There was no significant change in itchy sensation and corneal fluorescein staining. To the best of our knowledge, this is the first human clinical trial aimed at determining the influence of LF on AKC.

A randomized, double-blind, placebo-controlled clinical trials of LF had been previously published by Nakano [17] and Oda [18], evaluating the function of LF in human health. Herein, we have studied the effect of LF in a 12-week clinical trial.

Ono et al. had demonstrated the beneficial effects of eLF tablets (300 mg LF/d for 8 weeks) on visceral fat accumulation in Japanese men and women with abdominal obesity, and a tendency for greater improvement in the anthropometric data, compared with placebo [16]. Moreover, Kawakami et al. [3] studied the effects of eLF tablets (300 mg LF/d for 12 weeks) on the immune function and reported significant inter-group differences in peripheral blood lymphocyte subset ratios, neutrophils phagocytic function, and natural killer (NK) cell cytotoxicity. Therefore, for the present study, we selected an eLF dose of 400 mg, and a treatment period of three months, although future studies should increase treatment duration investigating dose-dependent patient responses. A higher LE dose has been reported to improve gingival health or skin conditions [17,18]. Neonatal infants ingesting 300–800 mg of LF per day from breast milk during the first few months of life [19]. Higher eLF might change LDH, IgE, TARC, and ECP serum levels.

Zimecki et al. [1] had reported the immunoregulatory effects of a nutritional preparation containing bovine lactoferrin, taken orally, while Sekine et al. [2] had reported the inhibition of initiation and early-stage development of aberrant crypt foci and enhanced NK cell activity in male rats that had been administered bovine lactoferrin concomitantly with azoxymethane. Pastori et al. had reported LF to reduce oxidative stress induced by tears of patients with keratoconus [20]. Previous studies have reported significant inter-group differences in peripheral blood lymphocyte subset ratios, neutrophils phagocytic function, and NK cell cytotoxicity [19,21,22,23,24,25], while the present study did not report any change in the number of eosinophils. The findings of previous studies implied that eLF administration might affect the human immune system [26], thereby reducing conjunctival injection in patients with AKC.

The possibility of LF interacting with the immune system has also been suggested [27,28,29,30]. In adults, LF is hydrolyzed by gastric pepsin, which prevents it from reaching the small intestine and binding to the LF receptors. Consequently, delivery systems, such as enteric-coated tablets, assure the physiological activity of LF in adults, since they protect it from gastric pepsin hydrolysis. The physiological actions of LF have been previously verified in human trials [31]. Previous studies have reported the immunoregulatory effects of LF in infants with low birth-weight [32], in patients with cancer [33], and in healthy adults [31]; however, only few studies have examined the possible benefits of LF during early life, for a normal healthy adult life, where an age-related loss of immune function is a concern. Our findings illustrate the potential use of LF for the prevention and treatment of allergic inflammation. Evaluation of eyelid skin elasticity was conducted using a cutometer, (model MPA580; Courage+Khazaka, Germany), with a 2-mm probe hole. Skin color was also measured using a portable reflectance spectrophotometer NF333 (Nippon Denshoku Industries Co. Ltd., Japan) (data not shown). No changes were observed in the atopic skin condition or bio-markers of allergic inflammation, which may be attributed to the low dose used in this study, short eLF treatment duration, or the influence of other mechanisms, such as innate lymphoid cells. Oda et al. had reported skin moisture to be greater in the 600 mg LF-treated group than in the placebo group [18]. Therefore, longer, and continuous studies are warranted in future, especially with higher LF doses. Recently, Biasibetti et al. have reported an improvement in atopic dermatitis in dogs [34], while Tong et al. reported the same in patients with dermatitis [35].

Corneal fluorescein staining showed no differences in these treatments. We assumed that the corneal condition had been already controlled using TALYMUS^®^, and thus LF had no effect on corneal inflammation. Itching did not significantly differ, implying that this synergic treatment had no effects on this sensation [36]. Notably, both itching and corneal fluorescein scores did not worsen due to LF treatment.

The current study had several limitations. We did not evaluate subjective symptoms using visual analog scale (VAS) assessment, nor did we assess the correlation between severity of subjective and objective symptoms for more accurate analyses. The current study did not exclude patients with dry eyes, which may have worsened their allergic changes [12,37], although few adolescent patients have itching, inherently, due to dry eyes [37,38].

In conclusion, significant differences in ocular allergic inflammation were observed between the LF with TALYMUS^®^ group and placebo group. Our results showed that, in patients with AKC, supplementation with LF may control certain aspects of atopic immune functions. Significant efficacy of the treatment was found in this study despite targeting TALYMUS^®^-treated patients with mild AKC. Moreover, the effects of TALYMUS^®^ and LF could continue, despite seasonal fluctuations. Further studies are warranted to elucidate the mechanisms underlying the allergic immune function of LF, as well as to examine the clinical importance of these changes. Overall, the current study suggested LF to potentially produce beneficial synergetic effects in patients with AKC, being treated with TALYMUS^®^.

## Figures and Tables

**Figure 1 jcm-09-03093-f001:**
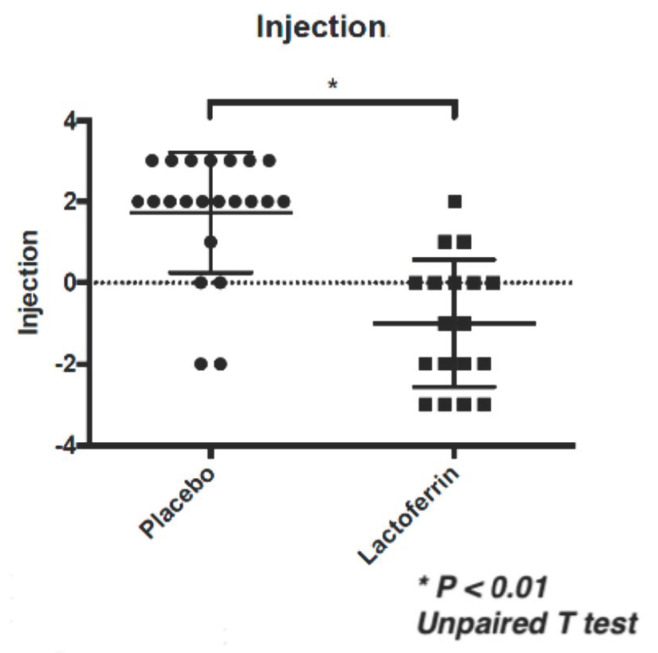
Conjunctival injection was significantly reduced in the lactoferrin (LF) with TALYMUS^®^ group than in the placebo group (12-week treatment). Both groups were also treated with 0.1% tacrolimus ophthalmic suspension. The injection was scored as follows; 0, 1, 2, and 3 in both the eyes. Horizontal bar represents the groups while the vertical bar represents the score; conjunctival injection score changed during the 12-week treatment (3 Month–0 Month). * *p* < 0.01, Unpaired T test.

**Figure 2 jcm-09-03093-f002:**
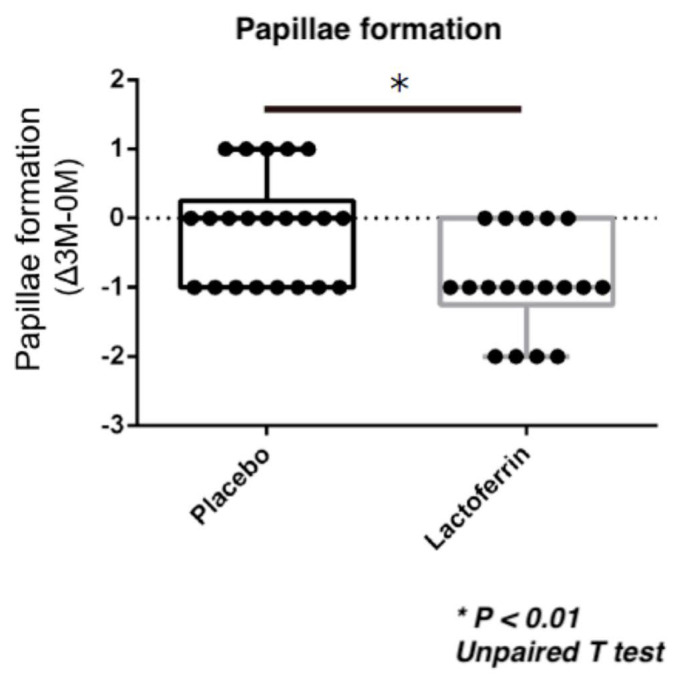
Conjunctival papillae formation in the enteric-coated LF (eLF) with TALYMUS^®^ group compared to that in the placebo group (12-week treatment). Both groups were also treated with 0.1% tacrolimus ophthalmic suspension. Papillae formation in the palpebral conjunctiva was evaluated using the following clinical evaluation criteria for allergic conjunctival diseases: diameter > 0.6 mm (3), diameter 0.3–0.5 mm (2), diameter 0.1–0.2 mm (1), and no manifestation (0). Horizontal bar represents the groups and vertical bar represents the score; papillae formation score changed over the 12-week treatment (Month–0 Month). * *p* < 0.01, Unpaired T test.

**Table 1 jcm-09-03093-t001:** Results of objective and subjective symptoms, pre- (0 weeks) and post-treatment (12 weeks).

		**0 M**			**3 M**			**⊿ (0 M–3 M)**			** p-value*		
											*0 M*	*3 M*	*⊿*
Itching	Placebo	0.8	±	0.7	1.0	±	0.7	−0.2	±	0.7	*0.73*	*0.49*	*0.24*
	eLF	0.9	±	0.6	0.9	±	0.8	0.0	±	0.5			
Injection	Placebo	1.3	±	1.2	3.0	±	1.8	1.7	±	1.5	*<0.01*	*<0.01*	*<0.01*
	eLF	2.6	±	1.7	1.6	±	1.6	−1.0	±	1.6			
Papillae	Placebo	1.3	±	0.6	1.2	±	0.7	0.1	±	0.8	*0.71*	*<0.01*	*<0.01*
	eLF	1.4	±	0.6	0.4	±	0.5	0.9	±	0.7			
Corneal stain	Placebo	0.2	±	0.4	0.5	±	0.7	−0.2	±	0.5	*0.97*	*0.21*	*0.17*
	eLF	0.2	±	0.4	0.2	±	0.4	0.0	±	0.5			
Data are shown by Mean ± SD. * Unpaired T test or Mann-Whitney test.													

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
