# Peer review of "Conjunctival Injection Reduction in Patients with Atopic Keratoconjunctivitis Due to Synergic Effect of Bovine Enteric-Coated Lactoferrin in 0.1% Tacrolimus Ophthalmic Suspension"

_jcm, 2020, doi:10.3390/jcm9103093_

Round 1
Reviewer 1 Report
The Article entitled “Conjunctival injection reduction in patients with atopic keratoconjunctivitis due to synergic effect of enteric-coated lactoferrin and 0.1% tacrolimus ophthalmic suspension” aim to determine the efficacy of the potential synergic effect of enteric-coated LF tablets (eLF) administered beside with ocular topical applied TALYMUS® that might improve conjunctivitis condition in patients with atopic keratoconjunctivitis (AKC).
The article is interesting and well written. It gives clinical information about the synergic effect of orally administered eLF tablets with topically applied TALYMUS®. The attention of this article is focused on the potential improvement of AKC eye disease treatment using lactoferrin (LF), thus I suggest major improvements of this article in:
- Introduction: need to be improved information about LF. The authors did not even mention that LF is already present in tears. Moreover, some information about the already known mechanisms of action should be added.
I suggest improving with recently published articles in 2020, for e.g. “Lactoferrin is already known to be an important component of the non-specific defense against infections and excessive inflammation; it is present at high concentrations in mucosal secretions, such as tears and breast milk [Terreni E, Burgalassi S, Chetoni P, et al. Development and Characterization of a Novel Peptide-Loaded Antimicrobial Ocular Insert. Biomolecules. 2020;10(5):664. doi:10.3390/biom10050664]. More specifically, serum lactoferrin is produced by acinar cells in the lacrimal gland and possibly also from tear neutrophils during infection and inflammation. By binding iron, lactoferrin prevents the pathogen from obtaining sufficient iron, which it relies upon for growth [Sabra S, Agwa MM. Lactoferrin, a unique molecule with diverse therapeutical and nanotechnological applications. Int J Biol Macromol. 2020;164:1046-1060. doi:10.1016/j.ijbiomac.2020.07.167].
Moreover, it was shown that oral administration of bovine LF could stimulate both systemic and mucosal immune responses in vivo” [Hanstock HG, Edwards JP, Walsh NP. Tear Lactoferrin and Lysozyme as Clinically Relevant Biomarkers of Mucosal Immune Competence. Front Immunol. 2019;10:1178. doi:10.3389/fimmu.2019.01178]
I suggest also to read the following article:
Rageh AA, Ferrington DA, Roehrich H, et al. Lactoferrin Expression in Human and Murine Ocular Tissue. Curr Eye Res. 2016;41(7):883-889. doi:10.3109/02713683.2015.1075220
- Discussion:
Line 178-179: authors state “Higher eLF doses might result in changes in LDH, IgE, TARC, and ECP serum levels”, I suggest improving the discussion of this sentence and add some references about it.
Line 198-199: authors speak about “prevention”. I suggest to argument about this assumption.
Other minor corrections:
I suggest using “eLF” or “LF” because is confusing the reader in some cases.
Figures: In the description of the figure the authors use “LF”, is eLF or LF?
Table 1: I suggest reducing the space between Mean +/- SD to make it more readable; Moreover, I suggest again to use always the same acronym "eLF" or “LF”.
Author Response
The article is interesting and well written. It gives clinical information about the synergic effect of orally administered eLF tablets with topically applied TALYMUS®. The attention of this article is focused on the potential improvement of AKC eye disease treatment using lactoferrin (LF), thus I suggest major improvements of this article in:
- Introduction: need to be improved information about LF. The authors did not even mention that LF is already present in tears. Moreover, some information about the already known mechanisms of action should be added.
Thank you for your comment and suggestion. We have revised the manuscript accordingly and included more information in the introduction section to address your comment.
I suggest improving with recently published articles in 2020, for e.g. “Lactoferrin is already known to be an important component of the non-specific defense against infections and excessive inflammation; it is present at high concentrations in mucosal secretions, such as tears and breast milk [Terreni E, Burgalassi S, Chetoni P, et al. Development and Characterization of a Novel Peptide-Loaded Antimicrobial Ocular Insert. Biomolecules. 2020;10(5):664. doi:10.3390/biom10050664]. More specifically, serum lactoferrin is produced by acinar cells in the lacrimal gland and possibly also from tear neutrophils during infection and inflammation. By binding iron, lactoferrin prevents the pathogen from obtaining sufficient iron, which it relies upon for growth [Sabra S, Agwa MM. Lactoferrin, a unique molecule with diverse therapeutical and nanotechnological applications. Int J Biol Macromol. 2020;164:1046-1060. doi:10.1016/j.ijbiomac.2020.07.167].
Thank you for your suggestion. We have revised the text accordingly and included the references suggested
Moreover, it was shown that oral administration of bovine LF could stimulate both systemic and mucosal immune responses in vivo” [Hanstock HG, Edwards JP, Walsh NP. Tear Lactoferrin and Lysozyme as Clinically Relevant Biomarkers of Mucosal Immune Competence. Front Immunol. 2019;10:1178. doi:10.3389/fimmu.2019.01178]
I suggest also to read the following article:
Rageh AA, Ferrington DA, Roehrich H, et al. Lactoferrin Expression in Human and Murine Ocular Tissue. Curr Eye Res. 2016;41(7):883-889. doi:10.3109/02713683.2015.1075220
Thank you for your suggestion. We have revised the text accordingly and included the reference suggested
- Discussion:
Line 178-179: authors state “Higher eLF doses might result in changes in LDH, IgE, TARC, and ECP serum levels”, I suggest improving the discussion of this sentence and add some references about it.
Line 198-199: authors speak about “prevention”. I suggest to argument about this assumption.
Thank you for your suggestion. We have revised the text accordingly for better clarity.
Other minor corrections:
I suggest using “eLF” or “LF” because is confusing the reader in some cases.
Figures: In the description of the figure the authors use “LF”, is eLF or LF?
Table 1: I suggest reducing the space between Mean +/- SD to make it more readable; Moreover, I suggest again to use always the same acronym "eLF" or “LF”.
eLF refers to Enteric-coated LF. We have changed LF to eLF in some parts of the revised manuscript.

Reviewer 2 Report
SENT August, 28, 2020
RE: Manuscript review jcm-897132
The manuscript entitled “Conjunctival injection reduction in patients with atopic keratoconjunctivitis due to synergic effect of enteric-coated lactoferrin and 0.1% tacrolimus ophthalmic suspension” is a randomized double-blind placebo-controlled trial aiming to evaluate the possibility to enhance the therapeutic efficacy of 0.1% tacrolimus ophthalmic suspension with tablets of enteric-coated bovine lactoferrin, in the treatment of severe atopic keratoconjunctivitis.
The study topic is quite interesting and it is of clinical relevance considering that the atopic keratoconjunctivitis is a chronic ocular allergic disease that can induce a permanent damage of the ocular surface, leading to loss of vision, and that its pathogenesis and therapeutic strategies remain to be elucidated.
The study under review is well planned, presented and discussed.
Some minor issues need to be addressed, which include:
- Title: the fact that the Lactoferrin (LF) used in the study was of bovine origin could be specified;
- Introduction: in this section the authors should report that the Lactoferrin (LF) has been identified in the tears and vitreous humor; that the LF gene expression has been confirmed in the cornea and retinal pigment epithelium (RPE) cell cultures form humans, and that the antibody reaction for human LF has been identified in the cornea, iris and RPE tissues (Rageh AA, et al. Lactoferrin expression in human and murine ocular tissue. Curr Eye Res 2016; 41:883-889). The authors could also specify that the lactoferrin present in the tear film has anti-microbial and anti-oxidative properties and that the oxidative stress is involved in the ocular surface inflammation (Seen S and Tong L. Dry eye disease and oxidative stress. Acta Ophthalmol 2018; 96:e412-e420); and that the LF present in the tear film has been demonstrated to have a potential as biomarker of mucosal immune competence (Hanstock HG, Edwards JP, Walsh NP. Tear lactoferrin and lysozyme as clinically relevant biomarkers of mucosal immune competence. Front Immunol 2019;10:1178.);
- Results, page 3, line 130: the authors should specify that the conjunctival injection and the conjunctival papillae formation in the palpebral conjuctiva were significantly reduced in the eLF groups as compared with the placebo group after 12 weeks of treatment;
- Results, Figures 1 and 2: in the text of both figures it is not clear that both groups (placebo and lactoferrin groups) were also treated with the 0.1% tacrolimus ophthalmic suspension.

Author Response
- Title: the fact that the Lactoferrin (LF) used in the study was of bovine origin could be specified;
Thank you for your suggestion, we have revised the title accordingly.
- Introduction: in this section the authors should report that the Lactoferrin (LF) has been identified in the tears and vitreous humor; that the LF gene expression has been confirmed in the cornea and retinal pigment epithelium (RPE) cell cultures form humans, and that the antibody reaction for human LF has been identified in the cornea, iris and RPE tissues (Rageh AA, et al. Lactoferrin expression in human and murine ocular tissue. Curr Eye Res 2016; 41:883-889). The authors could also specify that the lactoferrin present in the tear film has anti-microbial and anti-oxidative properties and that the oxidative stress is involved in the ocular surface inflammation (Seen S and Tong L. Dry eye disease and oxidative stress. Acta Ophthalmol 2018; 96:e412-e420); and that the LF present in the tear film has been demonstrated to have a potential as biomarker of mucosal immune competence (Hanstock HG, Edwards JP, Walsh NP. Tear lactoferrin and lysozyme as clinically relevant biomarkers of mucosal immune competence. Front Immunol 2019;10:1178.);
Thank you for your suggestion. We have revised the text accordingly and included all references suggested.
- Results, page 3, line 130: the authors should specify that the conjunctival injection and the conjunctival papillae formation in the palpebral conjuctiva were significantly reduced in the eLF groups as compared with the placebo group after 12 weeks of treatment;
Thank you for your suggestion. We have added “after 12 weeks of treatment” to the Results section
- Results, Figures 1 and 2: in the text of both figures it is not clear that both groups (placebo and lactoferrin groups) were also treated with the 0.1% tacrolimus ophthalmic suspension.
Thank you for your suggestion. We have added the suggested statement to figure legends 1 and 2.

Round 2
Reviewer 1 Report
The Article entitled “Conjunctival injection reduction in patients with atopic keratoconjunctivitis due to synergic effect of enteric-coated lactoferrin and 0.1% tacrolimus ophthalmic suspension” aim to determine the efficacy of the potential synergic effect of bovine enteric-coated LF tablets (eLF) administered beside with ocular topical applied TALYMUS® that might improve conjunctivitis condition in patients with atopic keratoconjunctivitis (AKC).
The article is interesting and well written. It gives clinical information about the synergic effect of orally administered eLF tablets with topically applied TALYMUS®.